# Selective Oxidation of Crude Glycerol to Dihydroxyacetone in a Biphasic Photoreactor

**Alexander Luis Imbault** and **Ramin Farnood** *

Department of Chemical Engineering and Applied Chemistry, University of Toronto, Toronto, ON M5S 3E5, Canada; Alexander.imbault@mail.utoronto.ca
* Correspondence: ramin.farnood@utoronto.ca; Tel.: +01-416-946-7525

**Abstract:** In this paper, the first biphasic photoreactor was introduced and utilized for the conversion of glycerol to glyceraldehyde (GAD) and dihydroxyacetone (DHA) using water and ethyl acetate as dispersed (active) and continuous (inactive) phases, respectively. Increasing the ethyl acetate content in the reactor improved the DHA yield; however, the optimal DHA selectivity was obtained at an ethyl acetate to water ratio of 90:10 (vol/vol). Compared to a monophasic photoreactor containing only water and identical amounts of glycerol and photocatalyst, the biphasic reactor containing 90 vol % ethyl acetate increased the DHA yield by a factor of 2.9 (from 4.5% to 13%) and the concentration of DHA by approximately 14 times (from 0.08 mM to 1.1 mM) after 240 min. Additionally, photocatalytic conversion of crude glycerol extracted using a 90:10 (vol/vol) ethyl acetate-water mixture showed a similar DHA conversion and yield to that of pure glycerol.

**Keywords:** photoreactor; photocatalysis; two-phase; glycerol; solar-to-chemical; dihydroxyacetone

---

## 1. Introduction

Crude glycerol is a low-value by-product that is generated at a rate of 10:1 by weight in biodiesel production process [1]. Crude glycerol is typically disposed of by burning; however, the steady increase in the global biodiesel demand has caused a significant oversupply of this material and has triggered the search for novel methods to valorize this renewable resource to fuels and chemicals [2,3]. Photocatalytic Solar-to-Chemical (STC) processes are an important application of green chemistry that allows incoming solar radiation to generate value added chemicals or fuels. Utilizing waste streams as a feedstock is an ideal way to further increase the positive environmental impact while utilizing an economical feedstock [4–6].

$TiO_2$ is a common photocatalyst for advanced oxidation processes (AOP) and is active for near UV light (<388 nm) [7]. The photoactivation of $TiO_2$ generates reactive species in water, mainly superoxide ions ($O_2^{\bullet-}$) and hydroxyl radicals ($OH^{\bullet}$) which then react further [8]. This has been used extensively to degrade hazardous compounds in water and air systems as well as generate valorized products [9,10].

Glycerol can be converted to dihydroxyacetone (DHA), a valuable pharmaceutical, by several routes including hydrogenolysis, TEMPO oxidation [11], thermo− [12], electro− [12] and heterogeneous catalysis [2,11–13]. DHA is valuable as a cosmetic compound used to reversibly darken the top layer of skin and is also responsible for a sweet sensory quality in wine [11,14]. Industrially, glycerol is converted to DHA by biological means, however, the process necessitates the purification of glycerol [11]. An alternative method for DHA production from glycerol is via heterogeneous photocatalysis in a slurry reactor [15]. Performance of a slurry photoreactor depends not only on the type and amount of photocatalyst but also on the photoreactor design [8,10,16]. Enhancing light distribution in photoreactors is key to improving the overall reactor performance and achieve optimum scale-up for industrial applications [8].

In this research, we examined the performance of a novel biphasic photoreactor for enhancing DHA production from both 'pure' and biodiesel derived crude glycerol. The contribution of dispersed and continuous phases on the overall conversion of DHA were studied separately in order to better explain the enhanced DHA production.

## 2. Results and Discussion

### 2.1. Monophasic Photoreactor System

The photocatalytic conversion of glycerol in ethyl acetate was investigated and the results are shown in Figure 1. Although the consumption of glycerol seemed to follow a consistent first order decay, the generation of DHA and GAD appeared to peak after about 120 min and 180 min, respectively. This suggests a competition between the generation of these species and their further oxidation to other compounds such as glyceric and glycolic acid.

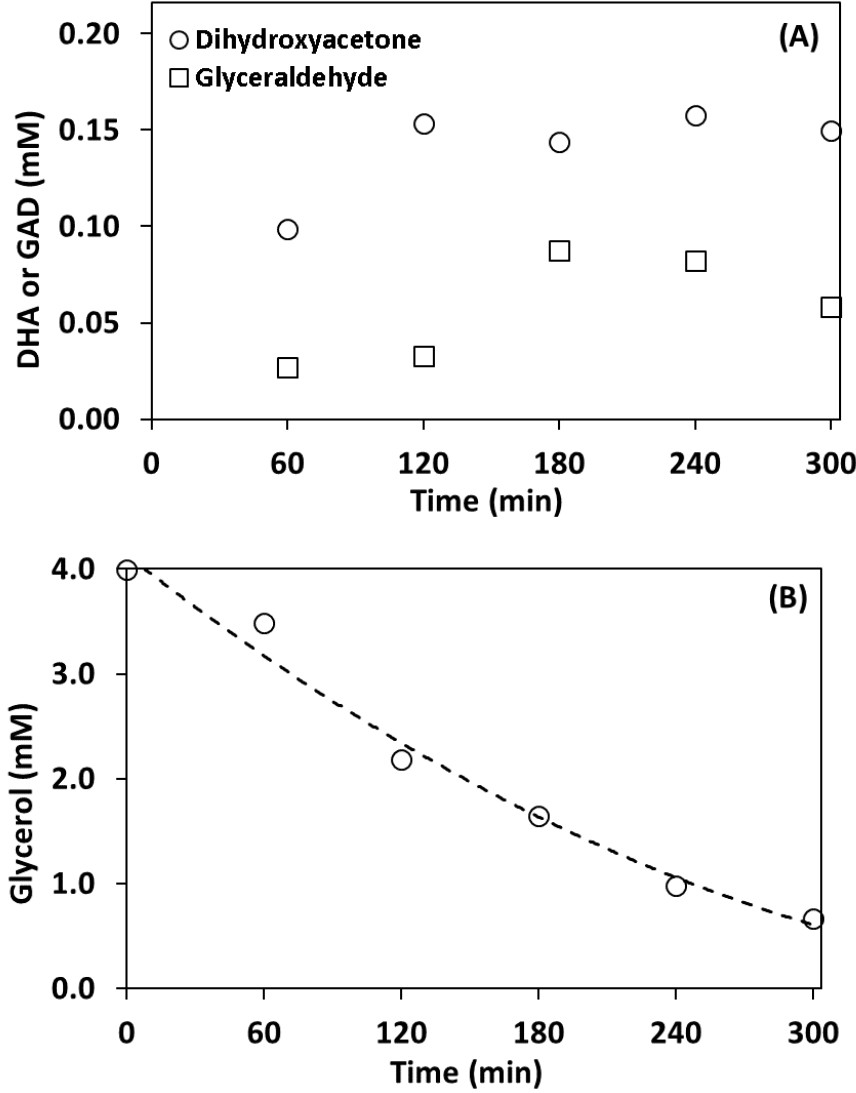

**Figure 1.** Photocatalytic reaction of glycerol in ethyl acetate under simulated solar light. (**A**) The concentration of dihydroxyacetone (DHA) and glyceraldehyde (GAD), and (**B**) concentration of glycerol over time. Ethyl acetate: 100 mL, glycerol: 0.4 mmol, Degussa P25 $TiO_2$: 0.1 g.

Comparing the glycerol degradation under identical conditions using water as the reaction medium (Figure S1), the half-life of glycerol in ethyl acetate was 45% less than that in water, namely

$t_{1/2}$ = 145 min versus 267 min. However, while both the monophasic ethyl acetate and water systems suffered from low DHA concentrations (i.e., in the order of 0.1 mM), the yield towards DHA was higher in ethyl acetate than in water, i.e., 8.7% versus 6.5%. Similarly, the maximum DHA:GAD ratio in ethyl acetate was 4.3 at 120 min which was an improvement over the identical reaction in water which attained its highest value of 1.7 at 300 min. The DHA:GAD ratio is a common metric in glycerol oxidation using photocatalysis and as it is important to maximize DHA production as the desirable product.

## 2.2. Biphasic Photoreactor System

Based on the above results, conversion of glycerol in ethyl acetate exhibited an improvement in DHA yield, however, still suffered from a low DHA concentration. To address this challenge, the conversion of glycerol in a biphasic reactor containing a mixture of water with ethyl acetate or hexane was examined. All concentrations reported for biphasic reactions were determined based on the volume of water (droplets) in the reactor.

## 2.3. Ethyl Acetate-Water Biphasic Photoreactor

In these experiments, 90 mL of ethyl acetate, 10 mL of water, 0.4 mmol of glycerol and 0.1 g of $TiO_2$ were added to the reactor. As seen in Figure S2, small droplets of water were formed and were dispersed in the ethyl acetate phase immediately upon mixing. These images also show that Degussa P25 $TiO_2$ particles strongly preferred water and hence, remined within the dispersed phase likely due to their hydrophilic nature. Based on Figure 2, DHA yield and concentration reached their respective peak values of 13% and 1.1 mM after 240 min irradiation that was significantly greater than their corresponding values in monophasic reactors.

The experimental results also show that the partition coefficient of glycerol between water and ethyl acetate was 450, while no DHA and GAD was detected in the ethyl acetate phase, indicating that these compounds were concentrated in the water (droplet) phase. Considering that $TiO_2$ was also preferentially retained in water, the biphasic reactor effectively composed of numerous small micro-reactors (water droplets) containing glycerol and $TiO_2$. The initial concentration of glycerol and the concentration of $TiO_2$ in these droplets were approximately 40 mM and 10 g/L, respectively, and photocatalytic conversion of glycerol occurred almost exclusively in the droplet phase.

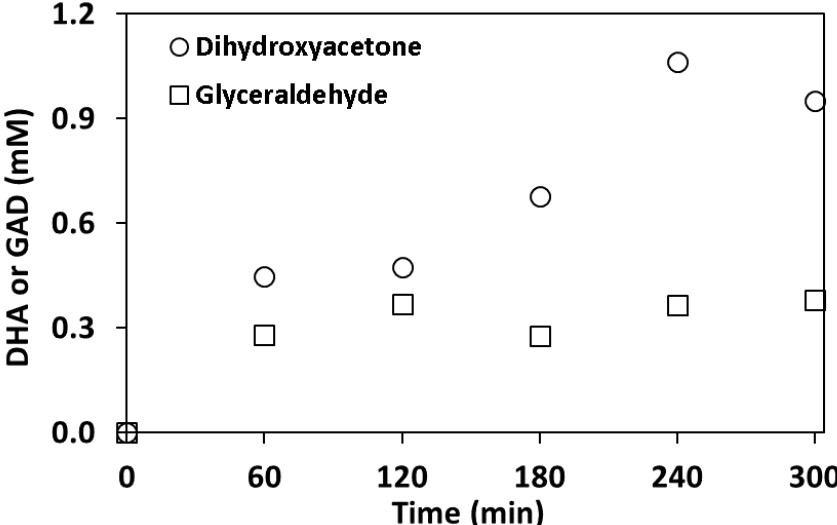

**Figure 2.** Photocatalytic reaction of glycerol in ethyl acetate and water under simulated solar light. The concentration of DHA and GAD in the water phase over time. Figure S3 shows a 360 min data point. Water: 10 mL, ethyl acetate: 90 mL, glycerol: 0.4 mmol, Degussa P25 $TiO_2$: 0.1 g.

Given that glycerol is soluble in water, its photocatalytic conversion mainly occurs in the water (droplet) phase. However, glycerol is slightly soluble in ethyl acetate such that 1 part of glycerol dissolves in 11 parts of ethyl acetate [17]. To determine the extent of glycerol conversion in the ethyl acetate phase in the biphasic reactor, control experiments were conducted as described earlier in the experimental section. The water removal results show a maximum DHA concentration of 0.24 mM after 180 min in the water phase, i.e., a maximum of about one-fourth of the DHA concentration in the biphasic reactor (Figure 2) could be attributed to the oxidation of glycerol dissolved in the ethyl acetate. However, glycerol conversion and DHA yield could not be determined with any accuracy since changes in glycerol concentration were less than the precision of the analytical methods used in this study.

Figure 3 illustrates DHA and GAD concentration in a monophasic photoreactor containing 40 mM glycerol solution in 100 mL water and 1 g TiO$_2$ under identical illumination conditions to that of Figure 2 to mimic the conditions in the water droplets. Although the DHA and GAD concentrations are higher than those reported for the monophasic reactor shown in Figure S1 (where glycerol concentration is 4 mM), they fall short of those of biphasic reactor. Using water saturated with ethyl acetate produced a similar result. The above findings suggest that the increase in the DHA and GAD concentrations in the biphasic reactor was likely due to the combination of two factors: (1) glycerol was preferably concentrated in water droplets and this higher concentration results in the improvement in glycerol conversion to DHA and GAD, and (2) dispersion of small water droplets in ethyl acetate (which has negligible light absorbance) resulted in a higher average light intensity within water droplets and more efficient light delivery to the photocatalyst than that in the monophasic reactor.

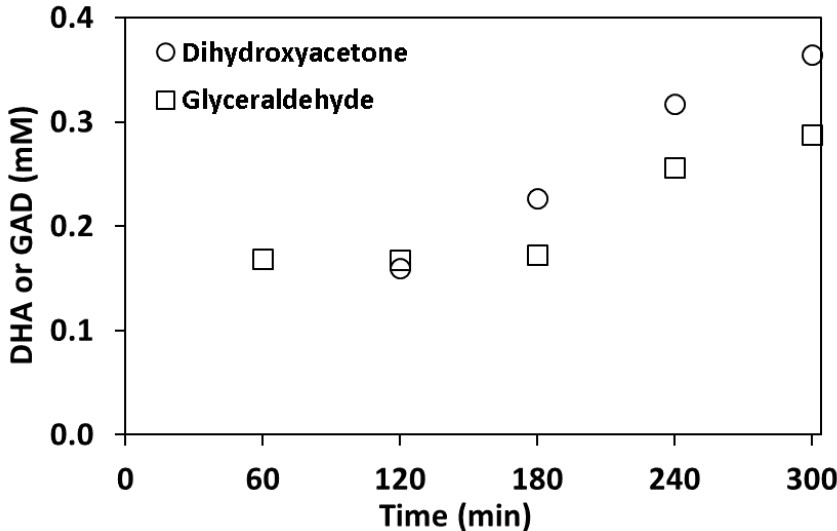

**Figure 3.** Photocatalytic reaction of glycerol in water under simulated solar light. The concentration of DHA and GAD in the water phase over time. Water: 100 mL, glycerol: 4 mmol, Degussa P25 TiO$_2$: 1 g.

### 2.4. Effect of TiO$_2$ Loading and Initial Glycerol Concentration

As discussed earlier, TiO$_2$ preferably concentrates in water droplets, creating 'high' concentrations of photocatalyst in the water phase of the biphasic reactor. A high TiO$_2$ concentration could hinder the light penetration within the droplets by light scattering and reduce the rate of glycerol conversion as well as DHA and GAD production. To examine the effect of TiO$_2$ concentration, an experiment was performed with a lower photocatalyst loading. Figure 4A shows that by decreasing TiO$_2$ loading by 10-fold to 0.1 g, DHA and GAD concentrations increased and peaked at 1.6 and 1.1 mM, respectively, after 180 min. When comparing the above results with reduced photocatalyst loading with those of ethyl acetate-water system in Figure 2, although the maximum DHA concentrations are similar the initial increase appears to be quicker. The decreased DHA:GAD ratio could indicate that GAD consumption is more sensitive to the change in photocatalyst loading than DHA production.

With the concentration of glycerol decreased by a factor of 10, it was fully consumed after 240 min (Figure S4B). Figure 4B shows that DHA and GAD concentrations peaked at 0.11 mM and 0.075 mM after 45 min and 90 min, respectively, and dropped to near zero after 240 min. Adsorption studies, as described in the experimental section, have shown that 19% of GAD and 12% of DHA were adsorbed by $TiO_2$, while no measurable adsorption of glycerol was detected on $TiO_2$ (even at a $TiO_2$ concentration of 10 g/L). Therefore, at low glycerol concentrations, DHA and GAD were preferably adsorbed on and degraded by $TiO_2$ likely to glyceric and glycolic acid then to $CO_2$ and $H_2$.

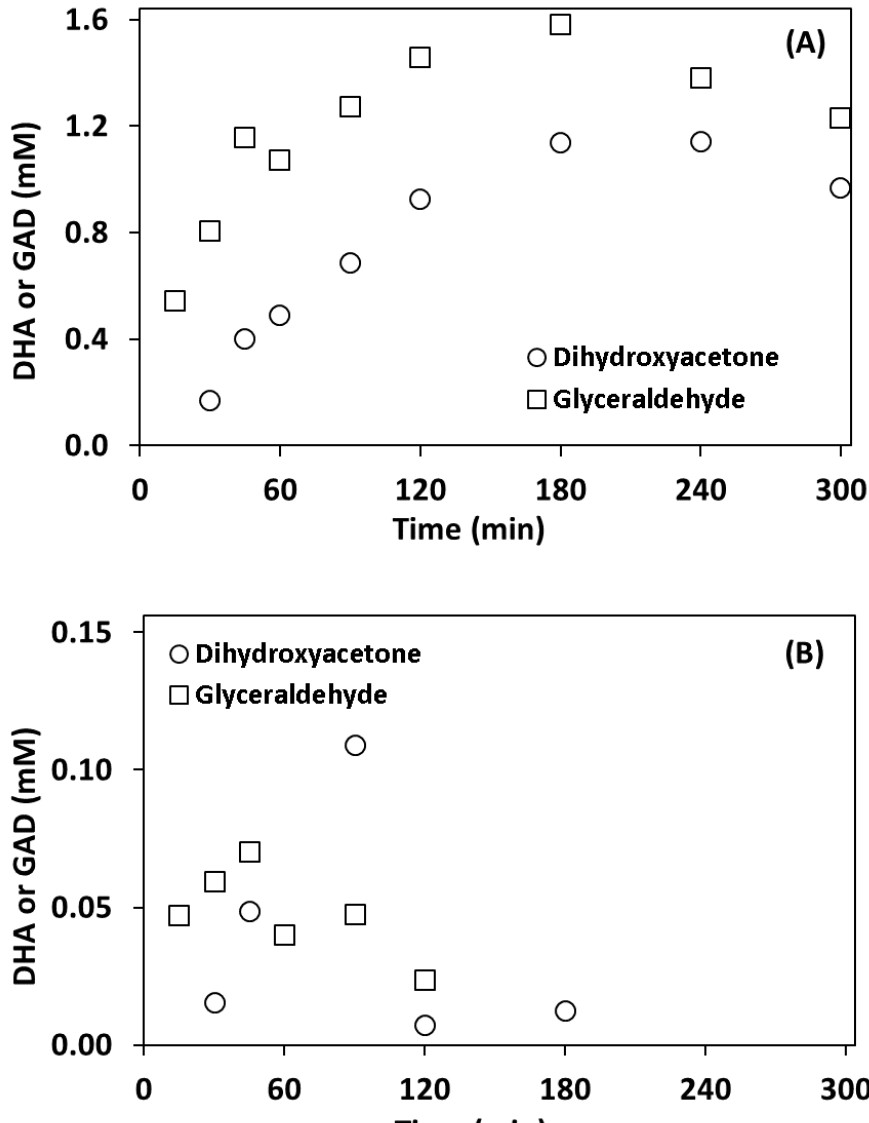

**Figure 4.** Effect of reduced photocatalyst loading on the concentration of DHA and GAD in the water phase: (**A**) glycerol loading of 0.4 mmol, and (**B**) glycerol loading of 0.04 mmol. Glycerol concentration over time is shown in Figure S4. Water: 10 mL, ethyl acetate: 90 mL, Degussa P25 $TiO_2$: 0.01 g.

*2.5. Effect of Ethyl Acetate to Water Ratio*

To determine the effect of ethyl acetate to water ratio on the performance of a biphasic reactor experiments were conducted at various ethyl acetate to water ratios. As seen from Table 1, DHA yield and concentration decreased by increasing the water (droplet) volume in the reactor. Given that the initial amount of glycerol remained constant in these experiments, as the volume fraction of water increased, the concentration of glycerol decreased that, as discussed earlier, could reduce the rate of

photocatalytic reaction. Moreover, increasing the amount of water (i.e., number of droplets) resulted in a greater droplet-droplet collision frequency and droplet coalescence rate. This led to a larger droplet size and hence lower average light intensity within the droplets and reduced reaction rates.

**Table 1.** Effect of ethyl acetate (EA) to water ratio on the performance of biphasic reactor after 240 min irradiation. Figure S5 contains the concentrations of DHA, GAD and glycerol over time. EA + water: 100 mL, glycerol loading: 0.4 mmol, $TiO_2$ loading: 0.1 g.

| EA:Water Ratio (vol:vol). | DHA Yield (%) | DHA Conc. (mM) | DHA:GAD Ratio |
| --- | --- | --- | --- |
| 95:5 | N.A. | 1.5 | 1.4 |
| 90:10 | 13 | 1.1 | 3.0 |
| 80:20 | 6.9 | 0.34 | 1.9 |
| 0:100 | 4.5 | 0.08 | 1.1 |

*2.6. Photocatalytic Conversion of Crude Glycerol*

In this experiment, photocatalytic conversion of crude glycerol extract (prepared as per the procedure described in the experimental section) was examined in a biphasic reactor. The reaction conditions were identical to those for 90:10 biphasic photoreactor using pure glycerol except for higher light intensity (light intensity was ~65% higher in crude glycerol experiments). According to Figure 5, after 300 min irradiation, DHA yield, DHA:GAD ratio and DHA concentration reached 14%, 0.70 and 1.1 mM, respectively. A sample was also collected after 24 h irradiation in which glycerol conversion was found to be 68%, however, the DHA and GAD concentrations were both significantly lower than those at 300 min (0.84 mM and 0.60 mM, respectively, as seen in Figure S6). The above DHA yield and concentration values were nearly identical to those of the 90:10 biphasic reactor with 'pure' glycerol after 240 min irradiation shown in Figure 2. However, given that the photoreaction rate constant varies proportional to the square root of light intensity [10] the corrected reaction time for pure glycerol would be 234 min ($300 \times \sqrt{1/1.65}$ min) that is practically the same as that of crude glycerol. This finding shows that crude glycerol, extracted using the procedure developed in this study, can be effectively converted to DHA.

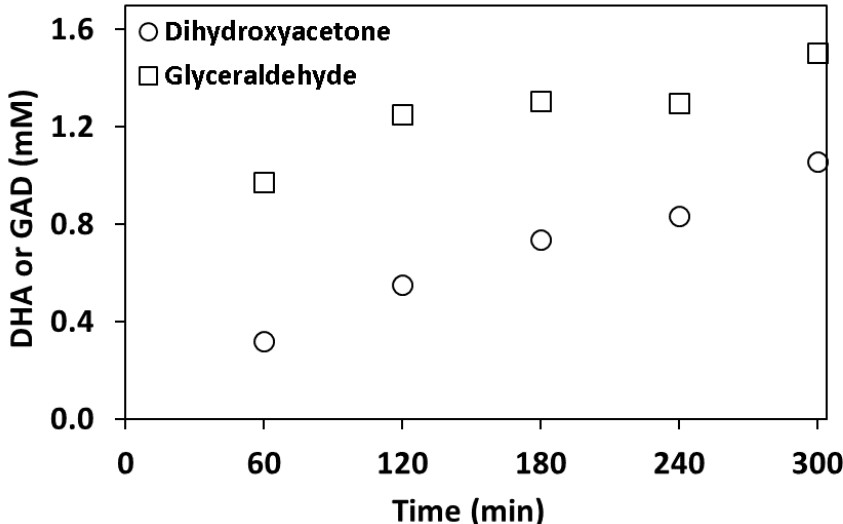

**Figure 5.** Photocatalytic reaction of crude glycerol in ethyl acetate and water under simulated solar light. Water: 10 mL, ethyl acetate: 90 mL, crude glycerol: 0.0622 g (~0.36 mmol of glycerol), Degussa P25 $TiO_2$: 0.1 g.

In our previous work, we reported conversion of crude glycerol to DHA using acetonitrile as a solvent. However, in that case, a sharp reduction in activity over time was observed that was attributed to the possible poisoning of the photocatalyst by impurities present in the crude glycerol [15]. In this

study, however, the biphasic reaction system did not suffer from such as reduced activity. GC-MS studies on the ethyl acetate and water phases separately, showed that methyl esters (a common impurity in crude glycerol) [18] were absent from the water phase and instead, were concentrated in the ethyl acetate phase. When only water was mixed with the crude glycerol under the same conditions described in the experimental section methyl esters were detected and glycerol was not extracted to a detectible degree (extraction was <0.1 mM from 0.05 g of crude glycerol). The removal of these impurities from the water phase by ethyl acetate was likely responsible for preserving the photocatalyst activity in a biphasic reactor. The similar polarities of methyl oleate and ethyl acetate shown in Table 2 explain why the water phase does not contain a detectable amount of methyl oleate.

**Table 2.** Relative Permittivity of relevant substances in the ethyl acetate–water–crude glycerol mixture.

| Compound | Relative Permittivity (T = 293K) | Ref. |
|---|---|---|
| Methyl Oleate * | 3.21 | [19] |
| Ethyl Acetate | 6.27 ** | [20] |
| Glycerol | 53.7 | [21] |
| Water | 80.2 | [22] |

* Common methyl ester found in biodiesel [18]. ** At 298 K.

### 2.7. Hexane-Water Biphasic Reactor

Based on the above findings, the biphasic photoreactor appears to have a superior performance compared to a monophasic reactor system. The improved performance of such a biphasic photoreactor is not limited to the choice of solvents and substrates used in this study and can be applied to a host of possible systems.

To illustrate this point, photocatalytic conversion of glycerol in a biphasic reactor containing 90 mL hexane, 10 mL water, 0.4 mmol glycerol and 0.1 g $TiO_2$ was examined. Figure 6 shows the concentration of DHA and GAD versus time in this system. No detectable concentration of glycerol, DHA or GAD was observed in the hexane phase.

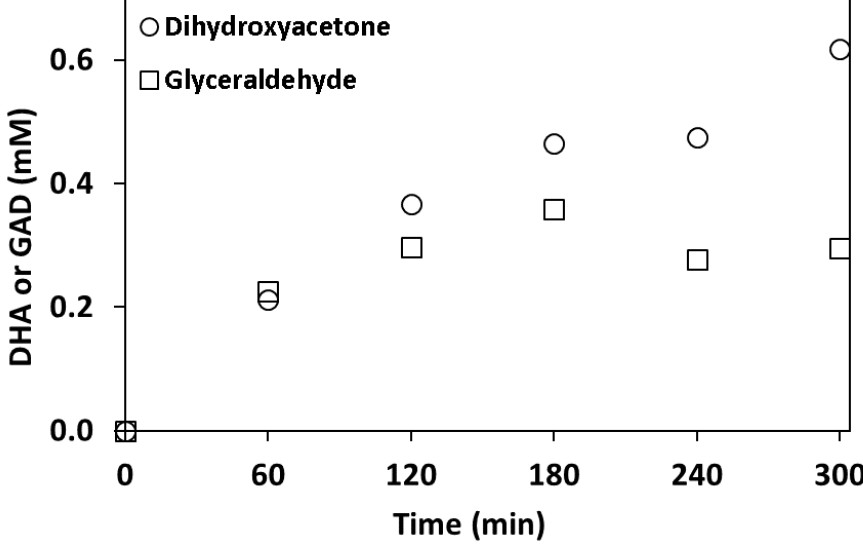

**Figure 6.** Photocatalytic reaction of glycerol in a mixture of hexane and water under simulated solar light. The concentration of DHA and GAD in the water phase over time. Glycerol concentration over time is shown in Figure S7. Water: 10 mL, hexane: 90 mL, glycerol: 0.4 mmol, Degussa P25 $TiO_2$: 0.1 g.

As can be seen in Figure 6, after 300 min of irradiation, the DHA:GAD ratio reached 2.1 while the yield towards DHA and glycerol conversion reached 5.8% and 36%, respectively. These values are

significantly higher than those observed for the monophasic photocatalytic conversion of glycerol in water, highlighting the benefits of such a biphasic photoreactor.

Compared to the ethyl acetate-water system (Figure 2), the DHA and GAD yields in a hexane-water system were lower. This could be attributed to the lower surface tension of water-ethyl acetate system (6.8 mN/m) than that of water-hexane system (50.2 mN/m). Additionally, the difference between the densities of hexane (0.655 g/mL) and water are significantly higher than that of ethyl acetate (0.894 g/mL) and water which notably caused mixing to be less efficient [23]. The combination of these factors likely resulted in a smaller average droplet size in water-ethyl acetate system, hence improved light intensity within the droplets and reaction yield.

## 3. Experimental

### 3.1. Materials and Chemicals

Degussa P25 Titanium dioxide and glycerol (>99%) were purchased from Sigma-Aldrich (Sigma-Aldrich, Mississauga, ON, Canada). Crude glycerol sample from biodiesel production was obtained from Biox Corporation (Biox Corporation, Hamilton, ON, Canada). Methyl orange (ACS reagent grade) was purchased from Fluka (Fluka, Oakville, ON, Canada). Ethyl acetate (HPLC grade), glycerol, glyceraldehyde (GAD), and dihydroxyacetone (DHA) were purchased from Fisher Chemical (Fisher Chemical, Markham, ON, Canada). Hexane was purchased from Caledon Laboratory Chemicals (Caledon Laboratory Chemicals, Halton Hills, ON, Canada). All chemical reagents were used without further purification. Ultrapure water (resistivity >18 MΩ·cm) obtained from a water purification system (Millipore, Etobicoke, ON, Canada) was used. 0.2 μm PTFE filters were purchased from VWR (VWR, Mississauga, ON, Canada).

### 3.2. Experimental Procedure

The photoreactor used was a custom blown 150 mL Pyrex glass batch photoreactor 80 mm in diameter and 53 mm in height equipped with an injection/sampling port and a quartz window (Φ36 mm) shown in Figure 7. The light source was a 1000 W solar simulator Xe arc lamp from Oriel Corporation (model 6269, see Figure S8).

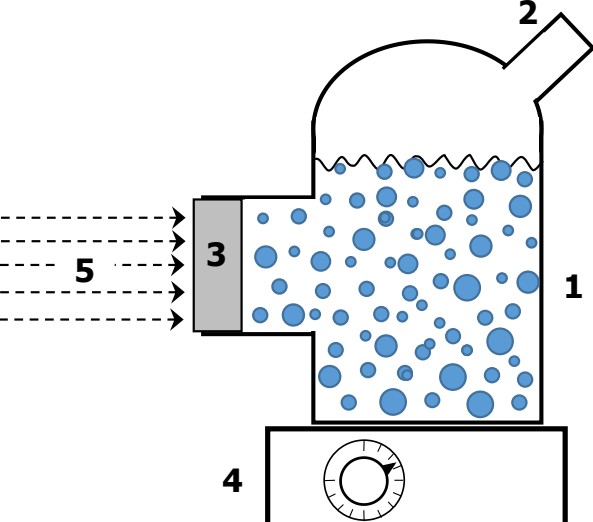

**Figure 7.** Schematic diagram of biphasic photoreactor showing continuous and dispersed phases. (**1**) Custom made photoreactor, (**2**) sampling port, (**3**) quartz window, (**4**) mixer, and (**5**) collimated beam of light from the xenon light source.

In a typical experiment, 100 mL of solvent, 0.1 g of Degussa P25 TiO$_2$ and 0.4 mmol of glycerol was added to the reactor. Reaction medium either consisted of a single solvent (i.e., water or ethyl acetate), or a mixture of water with ethyl acetate or hexane, in stated proportions. All experiments were performed at 25 ± 1 °C. All solutions were mixed in the dark for 15 min after which photocatalyst was added and mixed for another 15 min prior to illumination. All samples and standards were collected using a 0.2 μm PTFE syringe filter.

In crude glycerol experiments, a liquid–liquid extraction was performed to separate the glycerol and dissolve it into 90 mL of ethyl acetate and 10 mL of DI water. This had to be done due to the high viscosity of crude glycerol and the observation that without an extraction step it would not mix under the agitation used in the 'pure' glycerol experiments. This was done by adding a known mass of as received crude glycerol into a sealed polypropylene bottle along with 90 mL of ethyl acetate and 10 mL of DI water. The mixture was then placed in a VWR 1575 Incubator Shaker at 40 °C and 400 rpm for 24 h after which the aqueous extract phase was removed and used in the photoreactor.

To determine the extent of glycerol conversion in the ethyl acetate phase in the biphasic reactor, control experiments were conducted. In these experiments 90 mL ethyl acetate, 10 mL water, 0.1 g Degussa P25 titanium dioxide, and 0.4 mmol glycerol were mixed in the dark. Water and ethyl acetate were then separated by extracting the water phase was centrifuged for 10 min at 4500 rpm to reclaim the photocatalyst. The wet photocatalyst and ethyl acetate were then put back into the photoreactor and the reactor was subjected to irradiation in the absence of water. After the prescribed interval, the photoreactor was removed from the lamp's illumination, water phase was put back into the photoreactor, reactor was shaken vigorously to establish equilibrium between water and ethyl acetate phases, and samples were taken from the water phase. Water was again removed as described previously, and the above steps were repeated.

In addition, to observe the distribution of TiO$_2$ between the two phases in the photoreactor, high resolution images were captured using a Flea3 FL3-U3-88S2C high-speed digital camera from FLIR and a LM16JC10M 16 mm F1.8 Manual C-Mount lens from Kowa.

## 3.3. Adsorption and Partition Experiments

An adsorption experiment was conducted by mixing 100 mL of solvent (water or ethyl acetate) and 0.4 mmol of the solute (glycerol, DHA, or GAD) in a sealed 125 mL polypropylene bottle for 60 min and taking a 'before' sample. Next, 1 g/L of Degussa P25 TiO$_2$ was added to the mixture and stirred for another 24 h then an 'after' sample was taken. The amount of adsorption was then estimated from the ratio of the signal measured by HPLC-MS of the 'before' and 'after' samples.

Partition coefficient experiments were conducted by adding 90 mL of ethyl acetate, 10 mL of water and the solute of interest, either 0.4 mmol of glycerol. The mixture was stirred for 15 min before samples were taken from both ethyl acetate and water phases. The partition coefficient was calculated using

$$P_{EA/H2O} = C_{EA}/C_{H_2O}$$

## 3.4. Analytical Methods

The concentrations of glycerol, DHA and GAD were determined using HPLC-MS (Thermo Q-Exactive MS HESI II Dionex Ultimate 3000 UHPLC, ThermoFisher, Waltham, MA, USA) with a Phenomenex Luna-NH2 column (150 mm × 2 mm × 3 μm). The determination was carried out at 40 °C under constant flow rate of 0.3 μL/min and injection volume of 10 μL, mobile phase A, 0.1% formic acid in water mobile phase B, 0.1% formic acid in acetonitrile, 20:80 A:B for 3 min, 20:80 to 80:20 over 3 min, 80:20 for 3 min, 80:20 to 20:80 for 1 min and equilibrate for 5 min.

Glycerol conversion and DHA yield were determined according to

$$\text{Conversion}(\%) = \frac{C_0 - C_t}{C_0} \times 100$$

$$\text{Yield (\%)} = \frac{C_{DHA}}{C_0 - C_t} \times 100$$

where $C_0$ is the initial concentration of glycerol, $C_{DHA}$ and $C_t$ are concentrations of DHA and glycerol at the sampling time, t, respectively. The concentration of methyl esters, a component of crude glycerol, in water and ethyl acetate were determined using GC-MS (Varian 3900 GC and a Varian 2100T MS, Varian, Palo Alto, CA, USA) using an Agilent Factor Four column (VF-5ms, 30 m × 0.25 mm × 0.25 μm). The water samples were first dried down using an Eppendorf Vacufuge plus centrifugal vacuum concentrator with 48 spaces for 1.5–2 mL tubes on aqueous vacuum settings and ambient temperature (25 °C). The dried water samples were then reconstituted in an identical volume of ethyl acetate. The determination was carried out with a temperature profile as follows: hold at 40 °C for 1 min, increase to 250 °C at 6 °C/min, increase to 325 °C at 5 °C /min and then hold at 325 °C for 15 min. The injection volume was 1 μL with helium gas following at 1mL/min, a split ratio of 1:50 and an injector temperature of 260 °C. The MS was run using electron ionization with an emission current of 10 μamps and a transfer line temperature of 275 °C scanning m/z ratios from 35–650. The detector was held at 140 °C during the determination.

Methyl orange degradation was used to assess the stability of the output of Xenon lamp. Methyl orange concentration was determined using a UV-Vis spectrophotometer (Hitachi U-3900, Chiyoda, Tokyo, Japan) based on the absorption peak at 464 nm.

## 4. Conclusions

The proposed ethyl acetate-water biphasic reaction system for photocatalytic conversion of 'pure' glycerol resulted in the increase of yield towards DHA by a factor of 2 (6.5% to 13%) as well as enhancing the concentration of DHA by a factor of 18 (0.08 mM to 1.5 mM) after 240 min of reaction. Furthermore, three ethyl acetate:water ratios were investigated to better understand the system and how the water phase fraction could be adjusted to suit different needs such as yield, DHA:GAD ratio, DHA concentration or amount of DHA. It was found that although the 90:10 experiment yielded the highest amount of DHA and the highest DHA:GAD ratio (11 μmol and 3.0, respectively), the 95:5 experiment had the highest concentration of DHA (1.5 mM). A higher DHA concentration can facilitate the subsequent separation of DHA from the residual glycerol and other reaction products. The proposed process showed significant advantage in converting crude glycerol to DHA as the ethyl acetate was able to remove the impurities present in the crude glycerol and hence, prevent the poisoning of photocatalyst.

The improved performance of biphasic reactor may be attributed to two factors: firstly, dispersion of small water droplets in ethyl acetate likely resulted in a more efficient light delivery to the photocatalyst, and secondly glycerol was preferably concentrated in water droplets resulting in the improvement in glycerol conversion.

The biphasic reactor developed in this study could be applied widely to other photoreaction systems, independent of the choice of substrate and photocatalyst (or no photocatalyst) and may be extended to other immiscible solvent pairs.

**Supplementary Materials:** The following supplementary materials are available online at http://www.mdpi.com/2073-4344/10/4/360/s1, Figure S1: Photocatalytic reaction of glycerol in water under simulated solar light, Figure S2: Photographs of three mixtures of 0.1 g of Degussa P25 TiO$_2$, Figure S3: Photocatalytic reaction of glycerol in a mixture of ethyl acetate and water under simulated solar light, Figure S4: Effect of reduced photocatalyst loading on the concentration of glycerol in the water phase, Figure S5: Photocatalytic reaction of glycerol in a mixture of ethyl acetate and water under simulated solar light, Figure S6: Photocatalytic reaction of crude glycerol in a mixture of ethyl acetate and water under simulated solar light, Figure S7: Photocatalytic reaction of glycerol in a mixture of hexane and water under simulated solar light, Figure S8: Spectrum of the 1000W Xe UV enhanced bulb (6269).Video S1: Biphasic photoreactor.

**Author Contributions:** All authors have read and agree to the published version of the manuscript. Conceptualization, methodology, validation, formal analysis, investigation, data curation and writing, A.L.I. and R.F.; supervision, funding acquisition R.F.

**Funding:** This research was funded by NSERC, grant number RGPIN-2015-05982.

**Acknowledgments:** This work was supported by NSERC Discovery Grant funded by the Canada Government. Authors wish to thank BIOX Corporation for providing crude glycerol sample, Aaron Au for his help with photography, Goutham Rangarajan and Rohit Pal for their help with UV-Vis analysis as well as Robert Flick for method development and data analysis.

**Conflicts of Interest:** There are no conflicts of interest to declare.

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
