# Peer review of "Selective Oxidation of Crude Glycerol to Dihydroxyacetone in a Biphasic Photoreactor"

_catalysts, doi:10.3390/catal10040360_

Round 1
Reviewer 1 Report
Presented by Alexander Imbault and Ramin Farnood manuscript titled "Selective Oxidation of Crude Glycerol to Dihydroxyacetone in a Biphasic Photoreactor" covers the vital problem of generation of new methods of obtaining of vulnerable chemicals. Manuscript needs to be revised. All my comments I put in attached pdf file. General comment is that the authors should check manuscript pdf file during submission as there is a lot of grammar, editorial mistakes.

Author Response
Response to Reviewer 1
We have corrected the typos highlighted in the attached PDF. The figures not linking properly was not present in our submitted PDF or word document, this is something we will take special attention to look for in our final proof.
Experimental
Comment: There is no need to place the reactor scheme as it's typical apparatus.
Reply: This custom-made reactor has a few features that we feel it is important to keep Figure 1 in order to make it clear to the reader what we are using. This would be particularly helpful for the reproducibility of our results by other researchers.
Comment: In what proportions?
Reply: We use various proportions of our solvents as is stated in each relevant section. I have added “in stated proportions” to make this more clear to the reader.
Comment: Were the analytes firstly extracted or injected into GC-MS directly with water phase?
Reply: This has been added to the GC-MS portion of the experimental “The water samples were first dried down using an Eppendorf Vacufuge plus centrifugal vacuum concentrator with 48 spaces for 1.5-2 mL tubes on aqueous vacuum settings and ambient temperature (25 oC). The dried water samples were then reconstituted in an identical volume of ethyl acetate.”
Results and Discussions
Comment: Something is missing
Reply: We revised the link and will watch for this issue in the final prints.
Comment: Please check the pdf file before submission as Fig. 2 is really difficult to read.
Reply: This is a different formatting than the PDF and word document we submitted. We will watch for this.
Comment: It is better to use concentration per volume
Reply: We have intentionally chosen to use masses as it is more clear what is being kept constant despite changing volumes/volume fractions of water.
Comment: What does it exactly mean? Please rewrite the sentence.
Reply: I have rewritten the sentence to read “However, while both the monophasic ethyl acetate and water systems suffered from low DHA concentrations (i.e. in the order of 0.1 mM), the yield towards DHA was higher in ethyl acetate than in water, i.e. 8.7 % versus 6.5 %.”
Comment: So what? Why this ratio is important? Please explain.
Reply: The DHA:GAD ratio is a common metric in glycerol oxidation using photocatalysis and as it is important to maximize DHA production as the desirable product. This is now clarified in the manuscript.
Comment: This should be not in figure caption.
Reply: For consistency we did not include this 360 min data point, however we do want interested readers to be able to find this data in Fig. S.3 because they may be curious, like we were, if the maximum in DHA concentration at 240 min is a real maximum.
Comment: This sentence must be rewritten
Reply: We have revised the sentence to read “However, glycerol is slightly soluble in ethyl acetate such that 1 part of glycerol dissolves in 11 parts of ethyl acetate [18]”
Comment: of what?
Reply: I have adjusted the sentence slightly to be more clear reading “Water removal results showed a maximum DHA concentration of 0.24 mM after 180 min in the water phase, i.e. a maximum of about one-fourth of the DHA concentration in the biphasic reactor (Fig. 3) could be attributed to the oxidation of glycerol dissolved in the ethyl acetate.”

Reviewer 2 Report
In this publication, the conversion of glycerol to glyceraldehyde and dihydroxyacetone was performed in photoreactor in biphasic system. As catalyst was used titanium dioxide and as substrate crude glycerol from biodiesel production was used. This publication seems to be within the scope of journal. However it needs several corrections to be more acceptable for publication.
- In introduction, in sentence: “Glycerol can be converted to dihydroxyacetone (DHA), a valuable pharmaceutical, by several routes including hydrogenolysis, TEMPO oxidation, thermo-, electro- and heterogeneous catalysis”, references should be placed after every method.
- Last sentence on page 8 is not finished.
- In whole manuscript is lack of references to relevant tables and figures. Reference can not be in form of link.
- In description of photoreactor is lack of information about wavelength and maximum light strength of used lamp.
- In analytical method, please add information about filling of the columns, kind of detectors used in HPLC and GC analyzes.
- Please add information about temperature of detector.
- Figure 2 and 3 are absolutely illegible. Please correct their quality.
- References should be formatted according to the instructions of the journal.
Author Response
Response to Reviewer 2
Comment: In introduction, in sentence: “Glycerol can be converted to dihydroxyacetone (DHA), a valuable pharmaceutical, by several routes including hydrogenolysis, TEMPO oxidation, thermo-, electro- and heterogeneous catalysis”, references should be placed after every method.
Reply: I have revised the sentence to read “Glycerol can be converted to dihydroxyacetone (DHA), a valuable pharmaceutical, by several routes including hydrogenolysis, TEMPO oxidation11, thermo-12, electro-12 and heterogeneous catalysis2, 11-13.”
Comment: Last sentence on page 8 is not finished.
Reply: The sentence was continued oddly on the top of page 9. This seems to be an odd formatting the of the page with the following figure. I have added a space afterwards which has caused the sentence to appear complete on page 8.
Comment: In whole manuscript is lack of references to relevant tables and figures. Reference can not be in form of link.
Reply: This was not the case in the submitted PDF or word document and appears to be related to the formatting with the journal but appears to have been fixed. We removed all links and replaced them with correct reference numberings.
Comment: In description of photoreactor is lack of information about wavelength and maximum light strength of used lamp.
Reply: We have stated the type of lamp used (Xenon arc lamp), manufacturer and power. The spectrum of this type of lamp is available from the manufacturer at https://www.newport.com/mam/celum/celum_assets/1000_w_Xe_Lamp_Irradiance_600w.gif?50 (6269).
Comment: In analytical method, please add information about filling of the columns, kind of detectors used in HPLC and GC analyzes.
Reply: Information regarding HPLC and GC analyses is already provided under the Analytical Methods section. We have listed the commercial columns (HPLC used Phenomenex Luna-NH2 column and GC used Agilent Factor Four column), what the mobile phase was (HPLC has a changing mix of formic acid/water/acetonitrile and GC has helium gas following at 1mL/min) and the commercial detectors (for HPLC used Thermo Q-Exactive MS and for GC used Varian 2100T MS).
Comment: Please add information about temperature of detector.
Reply: We have added this information for the GC-MS section in the experimental “The detector was held at 140 oC during the determination.”
Comment: Figure 2 and 3 are absolutely illegible. Please correct their quality.
Reply: Unfortunately, they were not in this state in the PDF and word documents submitted. These are corrected and we will keep an eye on this for future proofs.
Comment: References should be formatted according to the instructions of the journal.
Reply: This has been corrected to match the format given in the Catalysts instructions.
Additional corrections:
Authors performed an additional experiment looking at the glycerol concentration when only water was used to extract glycerol from crude glycerol that we feel is important to share with the readers. Accordingly, the following sentence is added to section 3.6 (Photocatalytic conversion of crude glycerol): “When only water was mixed with the crude glycerol under the same conditions described in the experimental section, methyl esters were detected and glycerol was not extracted to any detectible degree (i.e. concentration of glycerol in the extract was <0.1 mM).”